# Spinal MRI in Patients with Suspected Metastatic Spinal Cord Compression: A Quality Improvement Audit in a District General Hospital in Kent, UK

**DOI:** 10.3390/ijerph22030401

**Published:** 2025-03-10

**Authors:** Michel-Elie Bachour, Rukhshana Dina Rabbani, Mahmudul Rahat Hasan, Sumaya Akter, Premsai Chilakuluri, Soirindhri Banerjee, Aruni Ghose, Elisabet Sanchez, Temitayo Ahmadu, Vasileios Papadopoulos, Jennifer Teke, David Bamidele Olawade, Saak Victor Ovsepian, Stergios Boussios

**Affiliations:** 1Department of Medical Oncology, Medway NHS Foundation Trust, Gillingham ME7 5NY, UK; michel-elie.bachour@nhs.net (M.-E.B.); rukhshana.rabbani@nhs.net (R.D.R.); sumaya.akter@nhs.net (S.A.); ch.premsai98@gmail.com (P.C.); soirindhribanerjee@yahoo.co.in (S.B.); aruni.ghose1@gmail.com (A.G.); elisabet.sanchez@nhs.net (E.S.); 2Department of Acute Medicine, Medway NHS Foundation Trust, Gillingham ME7 5NY, UK; mahmudul.hasan@nhs.net; 3Department of Research and Innovation, Medway NHS Foundation Trust, Gillingham ME7 5NY, UK; j.teke@nhs.net (J.T.); david.olawade@nhs.net (D.B.O.); 4Kent Medway Medical School, University of Kent, Canterbury CT2 7LX, UK; t.ahmadu2@kmms.ac.uk; 5Department of Urology, Medway NHS Foundation Trust, Gillingham ME7 5NY, UK; vpapadoster@gmail.com; 6Faculty of Medicine, Health and Social Care, Canterbury Christ Church University, Canterbury CT1 1QU, UK; 7Faculty of Engineering and Science, University of Greenwich London, Chatham Maritime ME4 4TB, UK; s.v.ovsepian@greenwich.ac.uk; 8Faculty of Medicine, Tbilisi State University, 0177 Tbilisi, Georgia; 9Faculty of Life Sciences & Medicine, School of Cancer & Pharmaceutical Sciences, King’s College London, London WC2R 2LS, UK; 10AELIA Organization, 9th Km Thessaloniki—Thermi, 57001 Thessaloniki, Greece

**Keywords:** metastatic spinal cord compression, magnetic resonance imaging, corticosteroids, decompressive surgery, palliative radiotherapy

## Abstract

Metastatic spinal cord compression (MSCC) is a common complication in cancer patients, occurring in 3–5% of diagnosed cases annually, and serves as the initial manifestation of malignancy in 20% of patients. Timely diagnosis and management are critical due to the risk of irreversible neurological damage and the significant impact on both quality and quantity of life. The National Institute for Health and Care Excellence (NICE) recommends that patients presenting with back pain accompanied by neurological signs and/or symptoms undergo whole-spine magnetic resonance imaging (MRI) within 24 h. This retrospective study at Medway Maritime Hospital in England aimed to assess adherence to these guidelines by reviewing the time from presentation to MRI for patients exhibiting symptoms and/or signs of MSCC. Data for 69 patients were collected over one year using electronic patient records and the acute oncology service database. Analysis revealed that MRI was conducted within 24 h in only 43 out of 69 cases (62%), and 16 out of 25 delayed cases (i.e., MRI done beyond the recommended 24 h window) experienced delays of more than 48 h. To improve guideline adherence, interventions such as informational flyers and regular MSCC training sessions, including trainee teaching and presentations during grand rounds, were implemented. A follow-up re-audit involving 113 patients over one year demonstrated improved adherence to the 24 h MRI guideline, with 81 out of 113 cases (71%) meeting the target. The second cycle also documented reasons for delays, identifying patient compliance and pain control as primary factors. Additionally, the timing of steroid administration following suspicion of MSCC was recorded. Future studies should re-assess adherence, focus on better documentation of delay causes, enhance pain management before MRI scans, and ensure prompt steroid administration.

## 1. Background

Metastatic spinal cord compression (MSCC) is a serious complication in cancer patients representing an oncological emergency. It is defined as the compression of the spinal cord or the cauda equina by metastatic tumour infiltration of the vertebral column leading to cord impingement by one or more of the following: epidural tumour growth, vertebral body collapse, or pathological fractures [1,2]. Due to the risk of irreversible neurological damage having a significant impact on both quality and quantity of life, timely diagnosis and management are essential [3]. It occurs in 5–10% of cancer cases annually [4] and is most frequently observed in patients with breast, lung, prostate, and kidney cancers, which collectively account for 60% of cases [5,6,7,8]. MSCC can also be the initial manifestation of malignancy in up to 20% of cases [9].

Clinically, MSCC can present with a variety of neurological symptoms and/or signs in addition to musculoskeletal involvement, the most common of which is back pain that can either be due to local bony destruction or radicular nerve compression. Motor dysfunction occurs when pyramidal tracts are involved, manifesting as weakness and gait disturbances, and can lead to complete paralysis. Practitioners can observe either upper or lower motor neuron signs depending on the onset of symptoms as well as the site of insult. Similar to motor signs, sensory tracts can be affected, causing paraesthesia and sensory deficits, and sensory-level and autonomic dysfunction (e.g., bladder and bowel incontinence or retention), which is a late-stage manifestation [10].

Diagnosing MSCC can be challenging and requires a low threshold of suspicion, which should be prompted especially in known cancer patients who are presenting with one of more of the signs and symptoms mentioned earlier. Following a clinical assessment, which consists of history taking and examination, clinicians proceed to obtaining radiological confirmation. The gold standard in diagnosing MSCC is whole-spine magnetic resonance imaging (MRI) with contrast; T2 sequences help in identifying the oedema as a sign of cord compression, while contrast-enhanced T1 images target the extent of metastatic cancer infiltration [11]. An alternative imaging modality in the absence of MRI (e.g., contraindicated), computed topography (CT) myelography, is recommended as a second line [2]. Other investigative measures that should be considered once the immediate treatment has been established as part of the general management plan include full-body staging CT, tissue diagnosis and biopsies, and tumour markers.

The acute management of MSCC should address multiple aspects simultaneously: steroid administration, symptomatic control, and a diagnostic approach.

For patients presenting with neurological deficits, prompt administration of dexamethasone is essential once MSCC is suspected [12] to help reduce the oedema surrounding tumour deposits and hence improve symptoms. However, to avoid the risk of gastric ulcers associated with high-dose steroids, they should be administered alongside gastric protective agents such as a proton-pump inhibitor. Steroid doses should also be gradually weaned down to minimise abrupt withdrawal complications. Additionally, supportive management should cover analgesia, which must be optimised, bladder and bowel care (e.g., urinary catheter for retention, laxatives for constipation), and venous thrombo-embolism prophylaxis due to reduced mobility.

Outside of the acute management of MSCC, the primary goal of treatment is typically palliative, aimed at alleviating pain and maintaining or restoring physical and neurological function. The choice of treatment depends largely on factors such as prognosis, neurological status, and potential for recovery. Local treatment options include palliative radiotherapy (RT), surgical posterior decompression with or without instrumentation, or total en-bloc spondylectomy [13]. Rehabilitation should be considered for MSCC patients, even when life expectancy is limited [14].

The National Institute for Health and Care Excellence (NICE) recommends that patients presenting with back pain and neurological symptoms undergo whole-spine MRI within 24 h [15]. Upon review of the previous literature, we encountered scarcity in studies quantifying the time interval between MSCC suspicion and obtaining an MRI [16,17,18], and more noticeably, the absence of an up-to-date, comprehensive audit with more than one cycle to follow up on interventions [19].

This is why we decided to conduct this retrospective, single-centre study aiming to determine the time between whole-spine MRI and presentation with neurological signs and/or symptoms, and to ensure that the MRI is being carried out within the recommended window.

## 2. Methods

This audit is a retrospective study conducted at Medway Maritime Hospital and consisting of two full cycles. Cycle 1 included a sample size of 69 patients from January 2022 to January 2023, and cycle 2 was performed following the implementation of interventions (in the form of MSCC teaching to residents, grand round, presentation of cycle 1 results, information emails and flyers) and included 113 patients from February 2023 to January 2024. Data were collected from clinical paper notes, electronic patient records (EPRs), and the acute oncology service (AOS) database using the Formic online proforma. The inclusion criteria comprised cancer patients (with a previous or current diagnosis) presenting with symptoms suggestive of MSCC, including any of the following—progressive lower back pain, sensory loss in the limbs, motor loss in the limbs, bowel and/or urinary dysfunction, and saddle anaesthesia.

## 3. Results

In cycle 1, MSCC was confirmed in 37 out of 69 patients (54%) (Figure 1). MRI was performed within 24 h of presentation in 43 out of 69 cases (62%) (Figure 2). A delayed case refers to any case where MRI was conducted beyond the recommended 24 h window. Of the remaining 26 delayed cases, almost two-thirds (16 cases) experienced delays of more than 48 h (23%).

The most common treatment for confirmed cases was radiotherapy (62%), followed by best supportive care (23%), surgery (8%), and a combination of surgery and radiotherapy (2%) (Figure 3). The time from MRI to treatment was less than 24 h, in accordance with guidelines [15], in 31% of cases (Figure 4).

The most common presenting symptom among all patients was back pain (85%), followed by motor symptoms (47%), sensory symptoms (18%), and autonomic symptoms (11%) (Figure 5). The most common site of cancer was the thoracic region (75% of all confirmed MSCC cases), followed by the lumbosacral region (42%) and the cervical region (8%) (Table 1).

In cycle 2, MSCC was confirmed in 67 out of 113 patients (59%) (Figure 6). MRI was performed within 24 h of presentation in 81 out of 113 cases (72%) (Figure 7). Of the remaining 32 delayed cases, two-thirds (21 cases) experienced delays of more than 48 h.

The cause of delay was documented in only 16% of the delayed cases, with patient compliance and pain control being the primary reasons. Steroids were administered within 4 h of MSCC suspicion in 44% of cases. The most common treatment for confirmed cases was radiotherapy (61%), followed by best supportive care (31%) and surgery (8%). No cases received a combination of surgery and radiotherapy (Figure 8). The time from MRI to treatment was less than 24 h in 28% of cases (Figure 9).

The most common presenting symptom among all patients was back pain (81%), followed by motor symptoms (47%), sensory symptoms (23%), and autonomic symptoms (20%). In patients with a confirmed diagnosis of MSCC, the percentages of these symptoms were 79%, 55%, 25%, and 20%, respectively (Figure 10). The most common site of cancer was the thoracic region (55% of all confirmed MSCC cases), followed by the lumbosacral region (43%) and the cervical region (18%), as shown in Table 1.

## 4. Discussion

The primary reason for the urgency in diagnosing and managing MSCC is to preserve neurological function, thereby improving quality of life and life expectancy, which is the ultimate goal of oncological interventions. MRI of the spine is a non-invasive, gold-standard diagnostic tool for MSCC, with a sensitivity of 93% and a specificity of 97% [20]. The first cycle of this study included a sample of 69 cancer patients who presented to Medway Maritime Hospital, a district general hospital and secondary care centre, with symptoms suggestive of MSCC over a 12-month period from January 2022 to January 2023. Of these, 43 patients underwent MRI within 24 h of presentation (Figure 2), representing 62% adherence to the recommended guidelines [15] and local policy (Figure 11). The remaining 26 delayed cases were categorised into two groups—delays of 24 to 48 h (9 cases) and delays exceeding 48 h (16 cases).

The delay in diagnosing MSCC has been recognised and documented in the literature. However, upon reviewing recent studies conducted at Medway Maritime Hospital [19], we noticed the absence of an up-to-date, comprehensive audit with two cycles to follow up on interventions. This motivated us to proceed with the second cycle to evaluate the outcomes of the implemented interventions. The interventions included organised teaching sessions for residents and middle-grade doctors on the clinical presentation, diagnosis, and management of MSCC; a grand round presentation of our first-cycle MSCC audit; presentations at local audit forums; distribution of informational flyers in key emergency department clinical areas and doctors’ facilities; and email highlights sent to the medical director for distribution to relevant departments.

The second cycle, or re-audit, was conducted over 12 months from February 2023 to January 2024 and included a sample size of 113 patients, using the same inclusion criteria as the first cycle—patients with known cancer, whether active or in remission, presenting with signs or symptoms suggestive of MSCC. The results showed an improvement in adherence, with MRI of the whole spine performed within 24 h for 81 out of 113 cases (72%), up from 62% (Figure 7).

The cause of delay was not documented in the first cycle but was noted in the second. Given the high patient turnover and generally understaffed wards, a decline in documentation quality is somewhat expected, especially when factoring in technical difficulties and limited access to computers. However, the cause of delay was recorded in only 5 out of 32 delayed cases. Of these, four were patient-related, involving poor compliance due to pain management issues and claustrophobia, while one case was attributed to human factors: the patient’s ward was changed, and MRI porters were sent to the original ward. When they were unable to locate the patient immediately, no new porter request was promptly made.

MSCC can present as a series of symptoms, with lower back pain being the most common presentation, followed by motor, sensory, and autonomic symptoms, as reported in the literature [4]. We observed the same pattern in our patient cohort: among the 69 cases studied, 58 reported lower back pain, 33 reported motor symptoms, 13 reported sensory symptoms, and 8 reported autonomic dysfunction (Figure 5). Symptoms often overlapped, with most patients presenting with more than one symptom. The second cycle showed similar results, with lower back pain and motor, sensory, and autonomic symptoms reported in 53, 37, 17, and 14 of the confirmed cases, respectively. Among all patients presenting before a confirmed diagnosis, the numbers were 92, 54, 26, and 23, respectively (Figure 10). Additionally, the site of compression due to metastatic cancer had a similar distribution in both cycles, as shown in Table 1, with the thoracic region being the most common, followed by the lumbosacral and cervical regions.

According to NICE guidelines [15] and the local MSCC policy pathway (Figure 11), patients should receive treatment within 24 h of confirming the diagnosis. Initial management should begin as soon as MSCC is suspected and includes adequate pain control with analgesia, the administration of high-dose steroids if not contraindicated, spinal immobilisation until imaging is completed and neurosurgical advice is obtained, management of urinary and bowel symptoms, and chemical or mechanical prevention of venous thromboembolism. When applicable, treatment extends beyond initial management to include RT, decompressive spinal surgery, or a combination of both. If the risks and side effects of treatment outweigh the benefits, a best supportive care approach is adopted, focusing on symptom management.

Data showed that in both cycles, only about one-third of patients (31% in cycle 1 and 28% in cycle 2) received treatment beyond initial management within 24 h of diagnosis (Figure 4 and Figure 9). The delayed cases were categorised into two groups: delays of 24 to 48 h, which accounted for 3% and 28% of delayed cases in cycles 1 and 2, respectively, and delays of more than 48 h, which occurred in 66% of delayed cases in the first cycle but improved to 44% in the second cycle.

The delays between diagnosis and treatment can be better understood by noting that Medway Maritime Hospital does not offer in-house services for RT or neurosurgery. Radiotherapy is provided at Maidstone and Tunbridge Wells NHS Foundation Trust, located 12 miles away, while neurosurgery is performed at King’s College Hospital, 34 miles away.

Neurosurgical decompression and spinal stabilisation combined with RT have been shown to yield better outcomes than RT alone, particularly in terms of regaining motor function, pain relief, and 1-year survival [11,21,22,23]. However, a relatively small proportion of patients with confirmed MSCC receive this combination treatment. This is due to selection criteria favouring patients with good performance status and a prognosis of over 6 months [19], as well as potential complications that may outweigh the benefits, such as major bleeding, hospital-acquired infections like pneumonia, cerebrospinal fluid leaks, and pulmonary embolism [24]. These factors were reflected in our data, with only 1 out of 37 confirmed MSCC cases in cycle 1 and none of the 67 MSCC patients in cycle 2 receiving neurosurgical intervention with RT. Instead, most patients received RT alone, followed by best supportive care, and, lastly, surgery alone. These results are illustrated in Figure 3 and Figure 8.

As part of the initial management, high-dose steroids should be administered if not contraindicated, as they reduce oedema, improve overall prognosis, and help alleviate pain [25]. The literature recommends administering high-dose steroids within 12 h of diagnosing MSCC [12], while our trust’s guidelines suggest administering them as soon as MSCC is suspected. In practice, we have established a 4 h window between presentation and steroid administration, based on the requirement that any patient presenting to the emergency department must be referred within 4 h. Since suspicion of MSCC should arise early during presentation, steroids should be administered by the time the referral is made, whether by the emergency department physician during the initial assessment or based on advice from the medical team receiving the referral. Data collected in the second cycle measured compliance with the 4 h window, highlighting that 47 out of 106 patients (excluding cases where steroids were contraindicated or already administered in the community) received Dexamethasone within 4 h of presentation, accounting for 44% of all suspected MSCC cases.

The data reflect an overall sub-optimal adherence to research guidelines in the domains of diagnosis in addition to the immediate management of MSCC. Although the data do not confirm the reason behind the delays in conducting MRI and administrating steroids, human factors such as communication and patient compliance were identified as potential causes.

This study paves the way for future projects targeting newer aims such as investigating the cause of delays in obtaining imaging and administering steroids, improving the quality of documentation, and enhancing adherence to guidelines in receiving treatment following a confirmed MSCC diagnosis. In addition to new projects, re-establishing the initial goal and re-measuring compliance to current guidelines should be attempted, and we should try to obtain an updated systematic review of the diagnosis and management of MSCC.

## 5. Conclusions

MSCC is an oncological emergency that occurs frequently not only in patients with known cancer but also in those without a prior diagnosis of malignancy. Due to the irreversible nature of neurological damage and its impact on both the quality and quantity of life, prompt diagnosis and management are crucial. The preferred diagnostic modality is MRI of the whole spine, which should be performed within 24 h of presentation in suspected MSCC cases, as per guidelines. The primary objective of our study was to measure adherence to these recommendations. The results highlight areas for improvement in the first cycle and demonstrate progress in the second cycle following intervention. This underscores the need for enhanced teaching and increased awareness to improve knowledge of MSCC suspicion, diagnosis, and management. Additional observations include the timing from MRI-confirmed diagnosis to definitive treatment, the administration of steroids as part of initial management, and the documentation of reasons for MRI delays. Future studies should aim to remeasure adherence, improve documentation to identify the causes of diagnostic delays, and enhance the timely administration of steroids.

## Figures and Tables

**Figure 1 ijerph-22-00401-f001:**
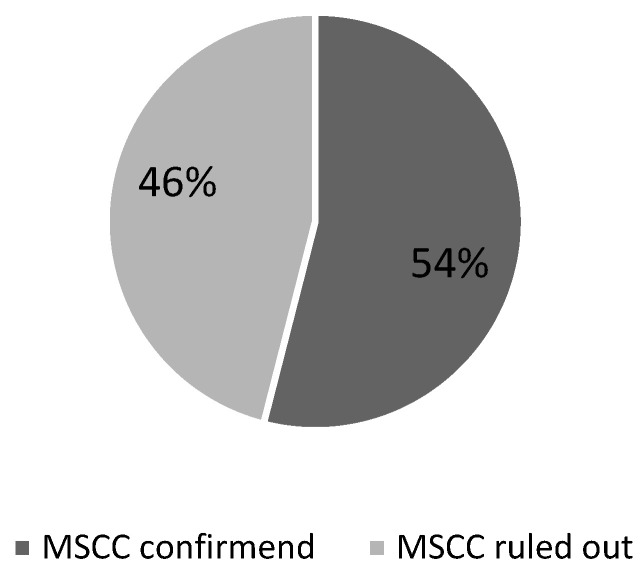
Confirmed MSCC amongst all suspected cases.

**Figure 2 ijerph-22-00401-f002:**
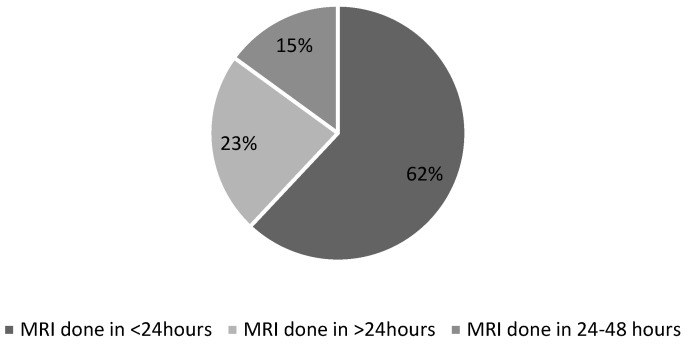
Distribution of patients based on time from presentation to MRI.

**Figure 3 ijerph-22-00401-f003:**
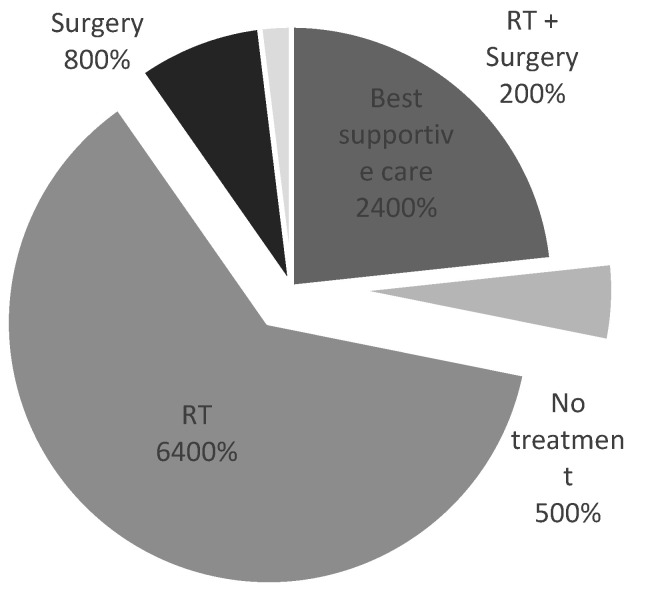
Types of treatment.

**Figure 4 ijerph-22-00401-f004:**
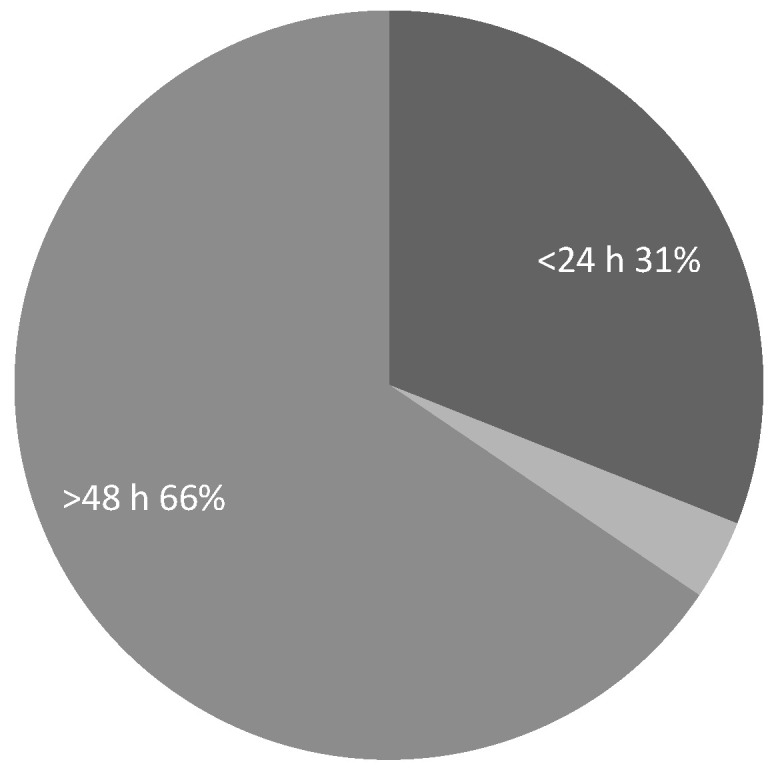
Distribution of patients based on time measured from MRI to treatment.

**Figure 5 ijerph-22-00401-f005:**
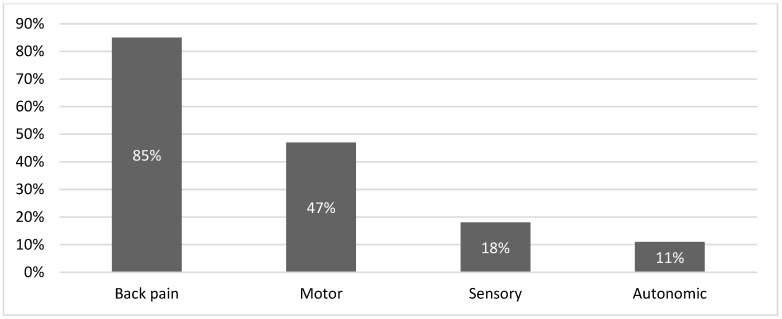
Presenting symptoms of all patients with suspected MSCC.

**Figure 6 ijerph-22-00401-f006:**
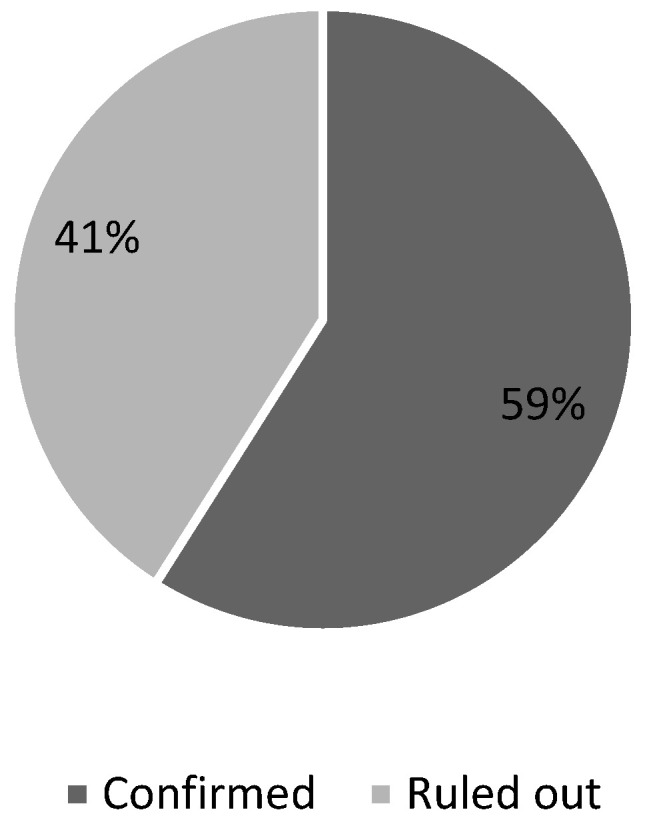
Distribution of confirmed cases amongst all cases of suspected MSCC.

**Figure 7 ijerph-22-00401-f007:**
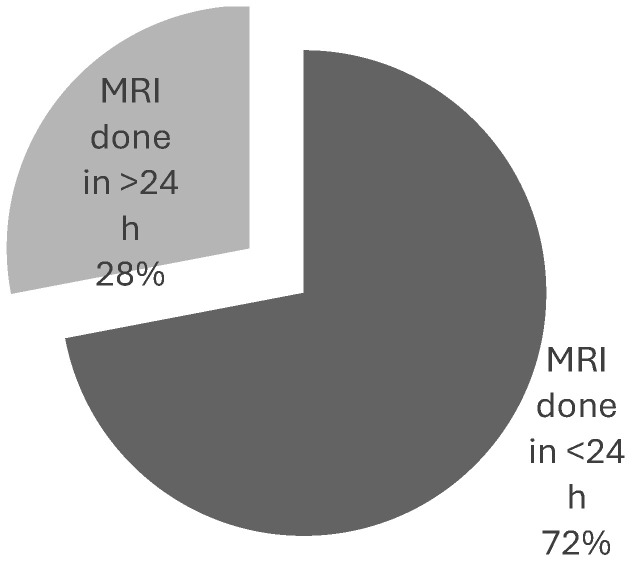
Distribution of patients based on time from presentation to MRI.

**Figure 8 ijerph-22-00401-f008:**
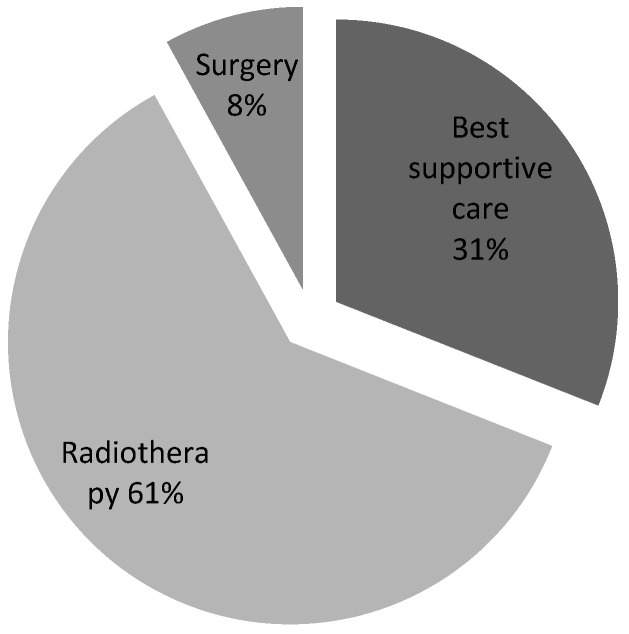
Types of treatment.

**Figure 9 ijerph-22-00401-f009:**
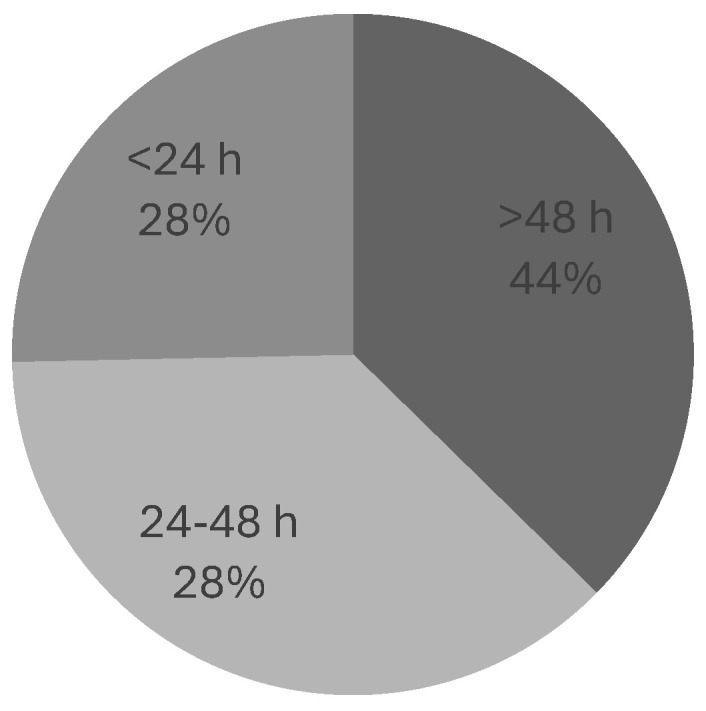
Distribution of patients based on time measured from MRI to treatment.

**Figure 10 ijerph-22-00401-f010:**
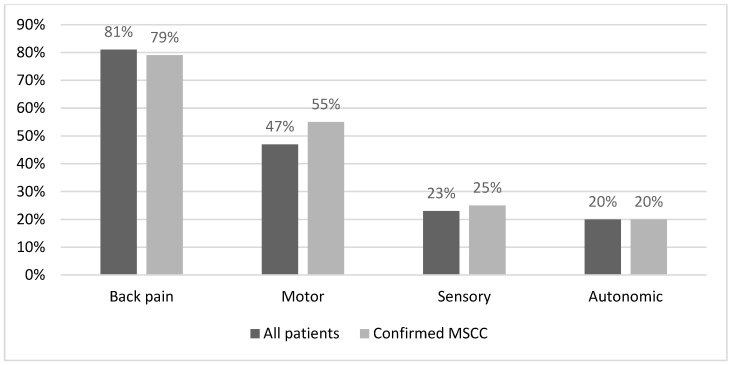
Presenting symptoms of all patients with suspected MSCC and confirmed MSCC.

**Figure 11 ijerph-22-00401-f011:**
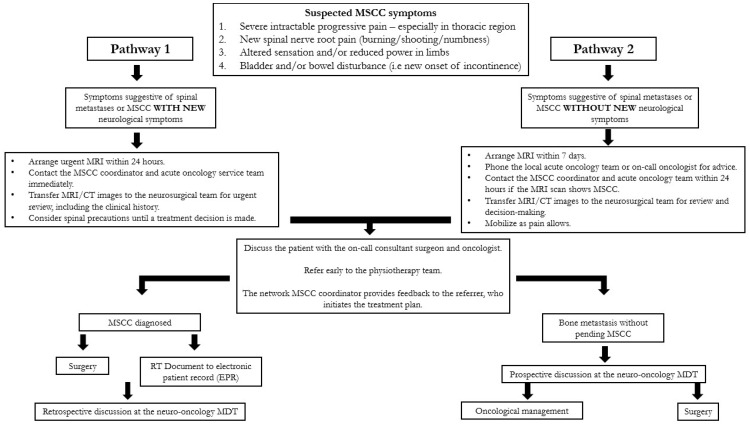
Medway Maritime Hospital updated MSCC guidelines. Abbreviations: MSCC: malignant spinal cord compression; MDT: multidisciplinary team; MRI: magnetic resonance imaging; CT: computerised tomography; RT: radiotherapy.

**Table 1 ijerph-22-00401-t001:** Site of cancer in patients with confirmed MSCC.

Site of Cancer	Number of Cases			
	Cycle 1		Cycle 2	
Cervical	3	8%	12	18%
Thoracic	28	75%	37	55%
Lumbo-sacral	16	42%	29	43%
Total confirmed MSCC cases	37		67	

## Data Availability

The data supporting the reported results can be found in the electronic patient report (EPR) and the acute oncology service (AOS) database in Medway Maritime Hospital.

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
