# Peer review of "Spinal MRI in Patients with Suspected Metastatic Spinal Cord Compression: A Quality Improvement Audit in a District General Hospital in Kent, UK"

_ijerph, 2025, doi:10.3390/ijerph22030401_

Round 1

Reviewer 1 Report

Comments and Suggestions for Authors

1- Please ensure that the hospital name is not mentioned in the title.

2- Please rewrite the Introduction section into 3-4 well-structured paragraphs. You may refer to the sources provided below to enhance clarity and depth

-Fried T, Foltz C, Lendner M, Vaccaro AR. How to Write an Effective Introduction. Clin Spine Surg. 2019 Apr;32(3):111-112. doi: 10.1097/BSD.0000000000000714. PMID: 30234565.

-Van Damme H. Steps to Writing an Effective Introduction. Acta Chir Belg. 2015 Jan-Feb;115:1. PMID: 26466390.

3-There is no need to capitalize the first letter of each word when writing 'Magnetic Resonance Imaging.

4-Could you please clarify how the definition of 'delayed case' was determined? Could you also provide a reference for this topic.

5-I don't understand the purpose of dividing the patients into two groups based on their development times in the last two years. Could you please clarify this.

6-Was ethical approval obtained and were informed consents taken from the patients?

7- The aim of the study is observational or descriptive, the absence of statistical analysis does not necessarily undermine the validity or originality of the work. However, if the research involves comparisons between different diagnostic techniques, treatment modalities, or patient outcomes, statistical analysis could help strengthen the study by providing quantitative evidence.

8-What outcomes were observed from performing early MRI on some patients and late MRI on others? Could you provide more details on the results section with statistical analysis.

Author Response

A document with the response to reviewer's comments has been uploaded.

Reviewer 2 Report

Comments and Suggestions for Authors

Dear authors, 

The paper addresses an interesting topic. 

The title is adequate. The case presentation is fluid and clear.

Schemes are well explained.

There are not images of spinal MRI.   

Author Response

(The authors gave the same response as above.)

Reviewer 3 Report

Comments and Suggestions for Authors

Thank you very much for the opportunity to review this very interesting article.

In their work, the authors investigate the adherence to NICE guidelines regarding MRI work-up in cases of suspected metastatic spinal cord compression, before and after a set course of action in a given medical centre.

Overall, it is a well-written article addressing a key topic. The reference list is up to date, the supplementary material is self-explanatory and the conclusions are well-supported by the results.

The authors should review Figure 3 (e.g. radiotherapy 6400%, surgery + RT 200%).

In the Methods section the authors mention the two time-frames from which the datasets were collected. Nevertheless it is only in the discussion section (lines 154-155) that the interventions made are mentioned. Possibly, an explicit description of the initiatives could be added in the methods section as well.

Investigating whether there is an actual statistical significance before and after the intervention would be advisable.

Author Response

(The authors gave the same response as above.)
